# Effect of Low-Temperature Plasma Carburization on Fretting Wear Behavior of AISI 316L Stainless Steel

Lu Sun [1,2], Yuandong Li [1,2,*], Chi Cao [1,2,3], Guangli Bi [1,2] and Xiaomei Luo [1,2]

[1] School of Materials Science and Engineering, Lanzhou University of Technology, Lanzhou 730050, China; sunhsyh12@163.com (L.S.); caochi@lut.edu.cn (C.C.); glbi@163.com (G.B.); lxmqingqing@163.com (X.L.)
[2] State Key Laboratory of Advanced Processing and Recycling of Nonferrous Metals, Lanzhou University of Technology, Lanzhou 730050, China
[3] Wenzhou Pump and Valve Engineering Research Institute, Lanzhou University of Technology, Wenzhou 325000, China
* Correspondence: liyd@lut.edu.cn

**Abstract:** AISI 316L stainless steel has received considerable attention as a common material for key ball valve components; however, its properties cannot be improved through traditional phase transformation, and fretting wears the contact interface between valve parts. A carburized layer was prepared on the surface of AISI 316L stainless steel by using double-glow low-temperature plasma carburization technology. This study reveals the effect of double-glow low-temperature plasma carburization technology on the fretting wear mechanism of AISI 316L steel under different normal loads and displacements. The fretting wear behavior and energy dissipation of the AISI 316L steel and the carburized layer were studied on an SRV-V fretting friction and wear machine with ball–plane contact. The wear mark morphology was analyzed by using scanning electron microscopy (SEM), the phase structure of the carburized layer was characterized with X-ray diffractometry (XRD), and the wear profile and wear volume were evaluated with laser confocal microscopy. The carburized layer contains a single Sc phase, a uniform and dense structure, and a metallurgically combined matrix. After plasma carburizing, the sample exhibited a maximum surface hardness of $897 \pm 18$ HV$_{0.2}$, which is approximately four times higher than that of the matrix ($273 \pm 33$ HV$_{0.2}$). Moreover, the surface roughness was approximately doubled. The wear depth, wear rate, and frictional dissipation energy coefficient of the carburized layer were significantly reduced by up to approximately an order of magnitude compared with the matrix, while the wear resistance and fretting wear stability of the carburized layer were significantly improved. Under different load conditions, the wear mechanism of the AISI 316L steel changed from adhesive wear and abrasive wear to adhesive wear, fatigue delamination, and abrasive wear. Meanwhile, the wear mechanism of the carburized layer changed from adhesive wear to adhesive wear and fatigue delamination, accompanied by a furrowing effect. Under variable displacement conditions, both the AISI 316L steel and carburized layer mainly exhibited adhesive wear and fatigue peeling. Oxygen elements accumulated in the wear marks of the AISI 316L steel and carburized layer, indicating oxidative wear. The fretting wear properties of the AISI 316L steel and carburized layer were determined using the coupled competition between mechanical factors and thermochemical factors. Low-temperature plasma carburization technology improved the stability of the fretting wear process and changed the fretting regime of the AISI 316L steel and could be considered as anti-wearing coatings of ball valves.

**Keywords:** AISI 316L stainless steel; low-temperature plasma carburization; carburized layer; fretting wear; wear mechanism

## 1. Introduction

AISI 316L austenitic stainless steel has excellent comprehensive mechanical properties (such as high strength, good corrosion resistance, low yield strength, good plasticity, and good toughness). Thus, AISI 316L steel is widely used in the petrochemical, biomedical,

aerospace, nuclear, and metallurgy industries [1–3]. However, AISI 316L steel also exhibits various disadvantages, including low hardness, poor wear resistance, and easy adhesion to abrasive parts. Moreover, the performance of AISI 316L steel cannot be improved through traditional phase transformation. These drawbacks significantly reduce the service life of AISI 316L stainless steel parts and affect their application range [4–6]. Hard sealing fixed ball valve stem/bearing, bearing/body, and spool/sealing surfaces are prone to fretting wear when the valve is normally open or normally closed. Fretting wears the contact interface between valve parts and accelerates the further initiation and expansion of the original cracks on the material surface. This results in the local failure of the bearings and spool, as well as the leakage failure of the sealing surface [7].

The wear resistance of various materials and engineering components can often be improved by surface modification. Double-glow low-temperature plasma carburization is an efficient thermochemical surface modification technology that can be used to increase the hardness and wear resistance of austenitic stainless steel [8,9]. For example, He et al. studied the fretting wear mechanism of 35CrMo steel before and after ion nitriding under dry friction and lubricated conditions. They reported that the wear resistance of the 35CrMo steel was significantly improved by ion nitriding. The high hardness and rough compound layer produced by ion nitriding were conducive to the formation of a third body layer at the fretting contact interface [10]. Adachi et al. treated AISI 316L stainless-steel-based tungsten carbide composites with single-plasma carburizing and continuous-plasma nitriding. Both the single carburizing and continuous processes can improve the surface hardness and corrosion resistance effectively [11].

Low-temperature plasma carburization (LTPC) technology is usually carried out at temperatures below 500 °C to avoid the precipitation of chromium carbides in the carburized layer. These chromium carbides reduce the corrosion resistance of stainless steel. LTPC enhances the hardness and wear resistance of austenitic stainless steel by forming a super-saturated austenite structure in the austenite structure (an Sc phase) and inducing a high level of residual stress [12–14]. Moreover, this strategy is inexpensive and has environmental benefits. Therefore, LTPC has been applied for the surface modification of austenitic stainless steels for several decades. Sun et al. prepared nitriding or carburizing layers on 316L stainless steel surfaces and studied the sliding friction of these surfaces under an ambient atmosphere and in corrosive media. Both layers provided the 316L steel with good wear resistance and corrosion resistance [15]. Montanari et al. treated 316L stainless steel with L-PBF and modified the steel surface with ion carburization. The results showed that ion carburization generated an Sc phase on the surfaces of the 316L steel and L-PBF-316L steel, which increased the surface hardness by a factor of two and further improved the mechanical properties [16]. Scheuer et al. prepared the carburized layer on the surface of AISI 420 martensitic stainless steel and investigated the effect of plasma carburizing time and temperature on their dry sliding friction and wear behavior. The carburized layers without Cr-carbide precipitation exhibited lower wear and friction coefficients. The wear mechanism of both samples exhibited micro-abrasion and oxidative wear [17]. Lamim et al. studied the influence of LTPC on the mechanical and tribological behaviors of sintered and nitriding sintered iron. Plasma carburizing the sintered iron surface effectively improved its mechanical properties. Moreover, the wear resistance was improved by approximately 45% [18].

Currently, studies investigating the wear behavior of austenitic stainless steel before and after plasma carburizing have mainly focused on evaluating the conventional reciprocating friction and wear behavior. However, little attention has been paid to the fretting wear behavior and mechanism of austenitic stainless steel after plasma carburization. Therefore, in this work, the influence of LTPC on the fretting wear behavior of AISI 316L austenitic stainless steel under varying loads and displacements was investigated using an SRV-V fretting friction and wear testing machine (Optimol Company, Germany). The morphology, phase structure, and surface hardness of the plasma carburization (PC) layers were thoroughly characterized. The wear morphology, elemental distribution, wear profile,

and friction coefficient were also analyzed, and the wear rate ($K$), cutting plastic ratio ($f_{cp}$), and frictional dissipation energy coefficient ($\alpha_e$) were calculated. In this study, the fretting wear behavior and mechanism of the AISI 316L austenitic stainless steel before and after LTPC are discussed in terms of mechanical and energy factors. This study provides experimental data support and a theoretical basis for evaluating and understanding the anti-fretting damage and surface protection of key components of ball valves.

## 2. Materials and Methods

### 2.1. Test Material and Carburized Layer Preparation

Commercial AISI 316L austenitic stainless steel ($\Phi$24 mm $\times$ 8 mm cylinder) was used as the lower sample; the chemical composition of this steel is shown in Table 1. The steel was pre-treated through grinding, cleaning, and drying until the surface was a smooth mirror. The corrosion solution was Marble reagent ($CuSO_4 \cdot 5H_2O$ (4 g) + HCl (20 mL) + $H_2O$ (20 mL)). The corrosion time was 10 s.

**Table 1.** The main chemical composition of AISI 316L stainless steel.

| Cr | Ni | Mo | Mn | Si | Fe |
|---|---|---|---|---|---|
| 16.45 | 10.01 | 2.1 | 0.92 | 0.36 | Bal |

The LTPC layer was prepared by treating the AISI 316L steel in an LDMC-30F glow ion diffusion furnace (Wenzhou, China). The specific process parameters are shown in Table 2. The sample was cleaned in the furnace for 20 min to remove surface stains before carburizing the surface.

**Table 2.** Detailed parameters for LTPC of AISI 316L stainless steel.

| Temperature (°C) | Voltage (V) | $H_2$ (L/min) | $C_2H_2$ (L/min) | Time (h) | Current (A) |
|---|---|---|---|---|---|
| 450 | 800 | 0.7 | 0.063–0.077 | 10 | 8 |

### 2.2. Fretting Friction and Wear Experiment

The fretting wear test was carried out with an SRV-V fretting friction and wear testing machine (Optimol Company, Germany). A GCr15 ball ($\Phi$10 mm) was selected as the upper sample, and the untreated or treated AISI 316L steel was used as the lower sample. The contact mode was ball–plane point contact. The experimental parameters are shown in Table 3. Before each test, the surfaces of the upper and lower samples were cleaned to ensure that the contact surface was clean. Each set of experiments was repeated at least three times, and the set with the most consistent results was selected for analysis.

**Table 3.** The experiment parameters of fretting wear test.

| Load (N) | Displacement (μm) | Frequency (Hz) | Time (min) | Temperature (°C) | Cycles |
|---|---|---|---|---|---|
| 30/50/70<br>50 | 70<br>50/75/100 | 25 | 20 | 25 | $3 \times 10^4$ |

### 2.3. Performance Testing and Characterization

The morphological structures of the carburized layer and the wear morphology marks were observed with field emission scanning electron microscopy (SEM, QUANTA FEG 450, FEI Company, Hillsboro, OR, USA) coupled with energy-dispersive X-ray spectroscopy (FEI Company, Hillsboro, OR, USA) to determine the elemental distribution. The morphological structures of the carburized layer were also observed with a metallographic microscope (Leica DMI 8, Leica Company, Wetzle, SBH, Germany). The phase structure

of the carburized layer was evaluated with X-ray diffractometry (XRD, D8-ADVANCE, Brooke Company, Germany). XRD patterns were obtained in the 2θ range of 20–90° with a step size of 0.02 (°)/s, and a copper target was used. The wear volume and wear profile were evaluated with an OLYMPUS OLS5000 3D measurement laser microscope (Olympus Corporation, Tokyo, Japan). The friction coefficient was automatically recorded by an SRV-V fretting friction and wear testing machine (Optimol Company, Germany). The surface hardness was measured using a microhardness tester (W1102D37, Buehler Company, Lake Bluff, IL, USA). The measurements were carried out using a 20 gf load applied for a duration of 15 s.

## 3. Results

### 3.1. Cross-Sectional Morphology and Properties of Carburized Layer

As shown in Figure 1a–c, the carburized layer was evenly distributed, continuous, dense, and metallurgically combined with the matrix. No obvious defects (cracks, holes, etc.) could be observed at the joint, and the boundary between the carburized layer and the matrix was clear [10,19]. The carburized layer mainly contained Fe, Cr, Ni, and C. The thickness of the carburized layer was approximately 25 μm. The surface hardness of the carburized layer was $897 \pm 18$ $HV_{0.2}$, which was approximately four times higher than that of the untreated AISI 316L steel ($273 \pm 33$ $HV_{0.2}$). The carburized layer had a surface roughness of 0.187, which is slightly higher than that of the untreated AISI 316L steel surface (~0.147). The increased surface roughness of the carburized layer compared to the untreated AISI 316L steel surface is due to the modified surface lattice of the austenitic stainless steel. During the LTPC process, a large number of carbon atoms were soldered in the tetrahedral or octahedral gaps of the austenite structure, which affected the lattice and induced the formation of an expanded austenite phase [9,20–22].

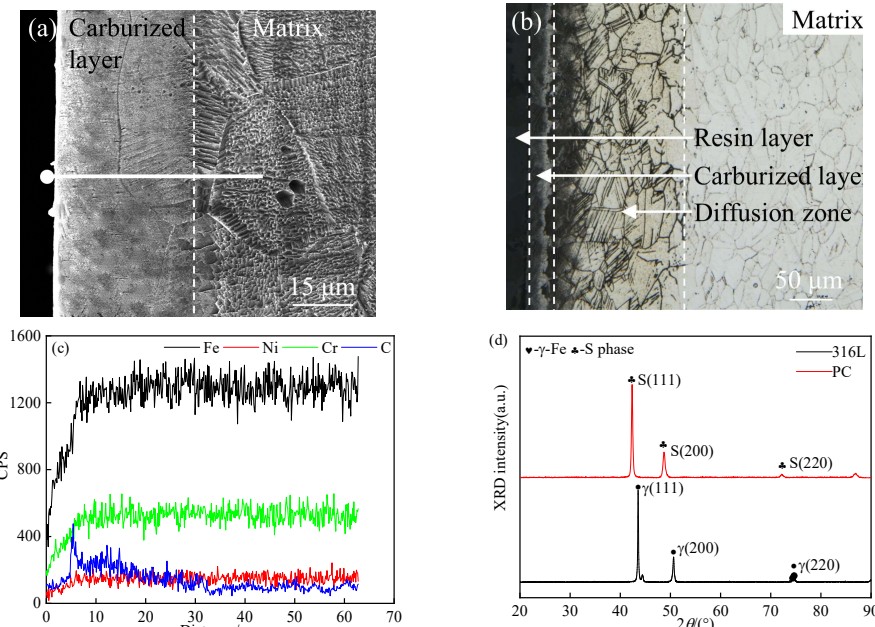

**Figure 1.** (**a**) Cross-sectional SEM microstructure of the carburized layer, (**b**) cross-sectional metallographic microstructure of the carburized layer, (**c**) EDS patterns of the carburized layer, and (**d**) XRD patterns of plasma-carburized AISI 316L steel.

The XRD analysis (see Figure 1d) showed that the carburized layer was composed of the Sc phase. Due to the strong affinity between the Cr atoms of the AISI 316L steel and the C atoms of the carburizing environment, the Cr atoms attracted a large number of C atoms to form chromium–carbon compounds during the LTPC process. However, the low temperatures used in this process did not provide enough energy to form chromium–carbon

compounds. Therefore, the C atoms entered the tetrahedral or octahedral gaps of the γ-Fe phase in a solid solution to form a metastable supersaturated solid-solution Sc phase. This phase had a disordered face-centered cubic lattice structure [9,23]. The formation of the carburized layer required the accumulation and reaction of stable C atoms, as well as the continuous diffusion and conversion of metastable C atoms.

After plasma carburizing, the main S (111) diffraction peak of the Sc phase shifted to a smaller angle compared with the main γ-Fe (111) diffraction peak of the matrix. Moreover, the S (111) diffraction peak was slightly broader after the LTPC process. Thus, the formation of the super-saturated solution of active carbon atoms in the AISI 316L steel austenite lattice gaps led to the expansion distortion of the original austenite lattice. Thus, the lattice constant changed. This lattice expansion was constrained by the internal matrix, resulting in residual stress in the superficial tangential direction. As a result, the modification of the crystal structure by the active carbon atoms caused textural changes, dislocation defects, slip band defects, lattice expansion, an increase in intergranular spacing, and higher deformation dislocation density. According to the Bragg equation ($2d\sin\theta = n\lambda$), when $n\lambda$ is constant, $\sin\theta$ decreases and d increases. Therefore, the shift in the main S (111) diffraction peak of the Sc phase to a smaller angle compared with the γ-Fe (111) peak of the matrix indicates larger d-spacing. The formation of the Sc phase and the coincidence of the diffraction peaks of the Sc phase and the primary γ-Fe phase caused the slight broadening of the main S (111) diffraction peak.

The enhanced surface hardness of the AISI 316L steel after plasma carburizing can be explained as follows [19,24–26]: (1) The carbon atoms have a strong solution-strengthening effect. During the LTPC process, a large number of C atoms continually diffused into the lattice gaps of the AISI 316L steel, resulting in solid-solution strengthening. This led to the rapid expansion distortion of the crystal lattice, increased the crystal surface spacing and lattice constant, and generated large residual stress. Thus, the surface hardness of the carburized AISI 316L steel was improved. (2) After plasma carburizing, the main diffraction peak of the Sc phase was broadened, indicating the existence of micro-strain in the carburized layer and the high defect density. This further improved the surface hardness of the AISI 316L steel after LTPC.

### 3.2. Friction Coefficient

The friction coefficient curves of GCr15/316L and GCr15/PC under different loads and displacements are shown in Figure 2. These curves are divided into three stages:

(1)  Initial stage I [27,28]: The friction coefficient curves of GCr15/316L and GCr15/PC sharply rose and very rapidly reached the maximum values. In the initial running-in stage, the friction pair experienced a slightly convex contact, the relative contact area was small, and fewer hard phase particles were shed, resulting in a low friction coefficient. With increasing load and displacement, the real contact area and roughness of the wear interface increased, the frictional heat of the wear surface increased, wear chips started to appear, the friction resistance increased, and the friction coefficient sharply rose.

(2)  Wear stage II [29,30]: The friction coefficient curve significantly decreased in this stage compared with the initial stage. The temperature between the contact surfaces continued to rise, and heat continued to accumulate. This resulted in local areas reaching the "friction flashover temperature". The thin oxide film formed on the wear surface slightly reduced the friction coefficient, but this oxide film was quickly crushed and peeled off by the sample on the wear surface. In addition, a large number of wear chips were generated and discharged, and the contact interface began to shift from two-body wear to three-body wear, resulting in fluctuations in the friction coefficient.

(3)  Stable stage III [31]: The friction coefficient curves were stable and showed an approximately straight line. Under variable load conditions, both the GCr15/316L and GCr15/PC friction coefficient curves required less time to enter the stable stage compared to variable displacement conditions. When the wear entered the middle and

late stages, the accumulation of wear particles between the contact surfaces led to the formation of a wear layer, and the contact interface completely shifted to three-body wear. At this point, the wear state remained relatively stable.

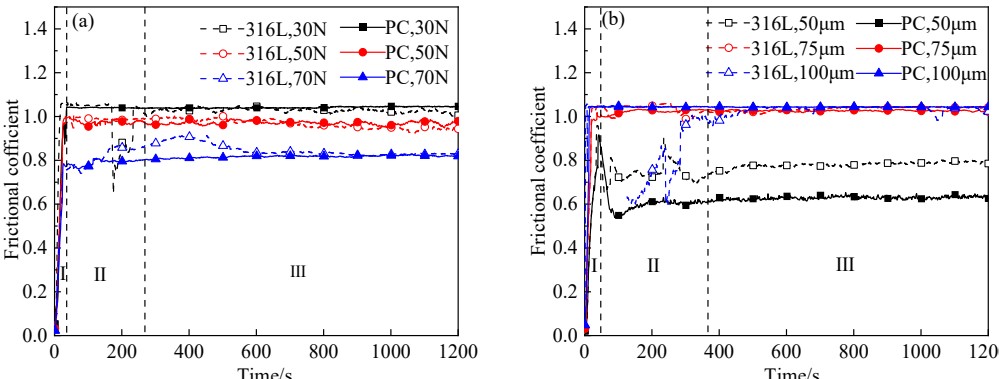

**Figure 2.** Friction coefficient curves of AISI 316L steel and carburized layer under different (**a**) loads and (**b**) displacements.

With increasing load, the friction coefficients of GCr15/316L and GCr15/PC decreased, and $\mu_{PC} > \mu_{316L}$ (see Figure 2a). Under a displacement of 50 μm, GCr15/316L and GCr15/PC exhibited their lowest friction coefficient values, and $\mu_{316L} > \mu_{PC}$. Under a displacement of 75 μm, the friction coefficients of GCr15/316L and GCr15/PC increased, and $\mu_{PC} > \mu_{316L}$. When the displacement was further increased to 100 μm, the friction coefficients of GCr15/316L and GCr15/PC did not increase, and the values were comparable to those obtained under a 75 μm displacement (see Figure 2b). The anti-friction effect of the carburized layer was not significant under varying loads and displacements.

The friction coefficient of GCr15/PC was slightly higher than that of GCr15/316L. There are four potential reasons that could explain this difference [32–34]: (1) The difference in the surface hardness of the AISI316L steel before and after carburizing. During fretting wear, abrasive chips participated in the wear process as a third body by scraping or ploughing the surface. The bearing capacity and anti-deformation capacity of these wear chips affected the friction coefficient. The influence of the relatively soft abrasive particles produced by the untreated AISI 316L steel on the friction coefficient was not as significant as that of the hard-phase abrasive particles produced by the carburized layer. (2) The rise in temperature due to friction at the wear interface during fretting wear softened the substrate surface more than the carburized layer. The friction coefficient of the AISI 316L steel was reduced to a certain extent because it was easier to form an oxide film than a carburized layer as the friction temperature raised. (3) The crystal structure of the steel sample also influenced the friction coefficient. Face-centered cubic austenite steel had a larger lattice than other crystal structures (BCC, HCP, etc.). The expansion distortion of the original austenite lattice after plasma carburizing meant that the carburized layer had a slightly larger friction coefficient than the untreated AISI 316L steel. (4) Surface roughness. The roughness of the carburized layer was slightly higher than that of the AISI 316L steel, and the number of micro-convex bodies on the contact surface was higher, resulting in higher relative shear strength and elevated friction stress. After the initial contact, the micro-convex bodies of the carburized layer were gradually sheared by the upper specimen, and the wear chips filled the contact interface. When the number of wear chips was less than enough to fill the micro-convex body troughs, the tribological behaviors increased such as extrusion, ploughing, and grinding. The friction coefficient increased. However, when the number of wear chips continued to increase to fill the micro-convex body troughs, the abrasive layer (the third body layer) could be formed at the contact interface and the friction coefficient decreased slightly. The third body layer could soon be extruded by ploughing and grinding the upper specimen to form new wear chips, and, therefore, the friction coefficient increased again.



The effects of load and displacement on the friction coefficient can be described in terms of mechanical and thermochemical factors. (1) Mechanical factors: Increasing the load increased the number of wear chips and intensified fatigue peeling. As more wear chips accumulated at the contact interface, the interface roughness increased. Moreover, the abrasive chips participated in wear by acting as "balls", increasing the friction coefficient. Increasing the displacement is conducive to the transverse expansion of wear, increasing the actual contact area, increasing the interface micro-convex body, and enhancing wear chip production. This results in an increase in tribological behaviors such as extrusion, ploughing, and grinding, leading to a higher friction coefficient. (2) Thermochemical factors: As the load and displacement increased, more heat accumulated at the contact interface. This softened the interface and reduced the shear strength of the material. At the contact interface, an oxide film was generated and sintered loose debris formed into a third body layer with a certain thickness, acting as a "cushion". Thus, the interfacial roughness was reduced, leading to a lower friction coefficient. Under variable load conditions, the thermochemical factors significantly influenced the friction coefficients of GCr15/316L and GCr15/PC, which decreased with increasing load. Under variable displacement conditions, the friction coefficients of GCr15/316L and GCr15/PC were mainly influenced by mechanical factors when the displacement was relatively small. However, with increasing displacement, the influence of thermochemical factors gradually became more significant. However, the influence of thermochemical factors was still limited compared with mechanical factors. Eventually, the friction coefficients of GCr15/316L and GCr15/PC stopped increasing with increasing displacement.

*3.3. Morphology and Composition of Wear Marks*

The wear marks of the AISI 316L steel presented as an irregular oval shape, as shown in Figure 3a,c,e. With increasing load, these wear marks gradually became round. The center areas of the wear marks showed serious adhesive wear, and localized peeling could be observed. Due to the low hardness and poor bearing capacity of the AISI 316L steel, adhesion and plastic deformation easily occurred in the wear marks. These wear marks were rough and irregular, and the fretting was in the slip regime at this time. After LTPC, the wear marks of the carburized AISI 316L steel showed a typical double-ring elliptic shape (see Figure 3d,f). The centers of these wear marks exhibited stickiness and spalling, the edges showed micro-sliding, parallel scratches were visible in the same direction as the movement of the upper sample, and the fretting was in the partial slip regime [22]. The carburized layer contained an Sc phase and high residual stress, and the surface hardness and strength of the AISI 316L steel were significantly increased after carburization. This reduced the adhesion effect and plastic deformation, and there was slight fretting wear.

A locally magnified view of the AISI 316L steel wear marks under a load of 30 N showed blocky and laminar wear chips stacked because of stacking faults (see Figure 3a). This is the third body layer. The edge of the abrasive layer was bright white, and this layer was surrounded by clusters of sintered or scattered abrasive particles. The compacted third body layer exhibited deformation and deep furrows in the same direction as the movement of the GCr15 balls, indicating adhesive wear and abrasive wear. The wear marks of the carburized layer partially consisted of the "scaly" third body layer formed by sintered granular debris and scattered abrasive particles, which appeared as adhesive wear. The local wear morphologies of the untreated AISI 316L steel and carburized layer under a load of 50 N showed several small wear pits and a large number of main cracks on the compacted surface (see Figure 3c,d). Around these main cracks, countless small micro-cracks sprouted and expanded, and local fatigue spalling and wear pits were observed. The scattered wear particles played a plowing role in the fretting wear, manifesting as adhesive wear, abrasive wear, and fatigue delamination. Under a load of 70 N, the wear morphologies of the AISI 316L steel and carburized layer showed a larger number of main cracks, and crack formation changed from a disordered to ordered pattern (see Figure 3e,f). Stronger fatigue wear could be observed: several parallel furrows and scattered abrasive particles were

present on the chip layer, the surface of the wear marks showed glaze characteristics, and there was serious plastic deformation. With increasing load, a large crack perpendicular to the movement direction of the friction pair gradually formed on the surface of the abrasion and was not worn away with increasing load. During the wear process, the wear rate was slower than the crack propagation rate, and local fatigue dominated the competition between local wear and fatigue [35,36]. Consequently, the wear mechanism was dominated by fatigue wear. With the increasing normal load, the wear mechanism of the AISI 316L steel changed from adhesive wear and abrasive wear to adhesive wear, abrasive wear, and fatigue delamination, and there was serious plastic deformation. Meanwhile, the wear mechanism of the carburized layer changed from adhesive wear to adhesive wear and fatigue delamination, and the furrow effect was serious.

Under variable displacement conditions, the wear morphology of the AISI 316L steel and carburized layer was elliptical, as shown in Figure 4. With increasing displacement, the AISI 316L steel wear marks increased in area, and the central adhesion became more serious. Meanwhile, the wear marks of the carburized layer showed a double-ring morphology with increasing displacement. At high displacements, the fretting regime of GCr15/316L was a slip regime, while that of GCr15/PC was a partial slip regime.

Under a displacement of 50 µm, a locally magnified view of the AISI 316L steel wear marks showed the presence of "cloud-like" convex block wear chips. The AISI 316L steel surface showed small and shallow wear pits and micro-cracks, while no obvious wear could be observed in the carburized layer (see Figure 4a,b). Under a displacement of 75 µm, the abrasive particles in the AISI 316L steel abrasive marks changed from a block to lamellar morphology, and fine sintered abrasive particles and micro-cracks were observed on the surface. Meanwhile, the carburized layer wear mark surface showed furrows, large wear pits, and flat abrasive particles (see Figure 4c,d). Under a displacement of 100 µm, the wear marks of the AISI 316L steel and the carburized layer were the third body layer composed of flat compacted wear particles. The existence of micro-cracks indicates that the material here will eventually be spalled (see Figure 4e,f). Because the friction pairs were always in close and nearly static contact during fretting wear, the wear interface can easily undergo serious plastic deformation due to the local high stress and rise in friction temperature. A large number of adhesion points were formed between the friction pairs. These adhesion points were then repeatedly pulled, sheared, broken, and compacted in the reciprocating motion of the upper sample. This repetitive process led to the appearance of wear marks such as furrows, fatigue delamination, material shedding, and wear pits. With increasing displacement, the wear mechanisms of both the AISI 316L steel and the carburized layer changed from adhesive wear to adhesive wear with slight fatigue delamination.

The worn-out surface showed two main forms of abrasive debris: (1) The wear chips were sintered and compacted to form a block or lamellar third body layer, which blocked the direct contact between friction pairs and acted like a "cushion" to slow down wear. (2) As the third body scattered and was embedded in the cracks, furrows, and wear pits, it acted like a "ball", which aggravated the wear process. The normal force compacted the hard-phase abrasive particles downward into the wear interface and increased the local contact stress of the contact interface. When the abrasive particles were pressed down until they reached a critical depth, the cracks began to grow and expand. There are two main forms of cracks on the ground surface: (1) longitudinal cracks, which were perpendicular to the direction of fretting wear and caused the initiation and propagation of micro-cracks, and (2) transverse cracks, which extended from the bottom of the pressed particles to the surface of the abrasion marks. When the transverse cracks converged and extended to the surface of the wear marks, fatigue shedding and the formation of new wear chips occurred on the surface. The upper specimen ball pushed the abrasive particles to plough (press in) or scratch (not press in) the wear interface under the action of tangential forces, thus forming furrows or scratches. This led to serious plastic deformation, which could be observed as abrasive wear and fatigue delamination. The carburized layer had a high hardness, a homogeneous structure, and good toughness. Thus, with increasing load and displacement,

the carburized layer performed better and had a solid lubrication effect during the fretting wear process. This reduced the adhesive wear and micro-crack formation compared to the unmodified AISI 316L steel, slowing down the fretting wear. Consequently, while the fretting regime of GCr15/316L was a slip regime, that of GCr15/PC was a partial slip regime.

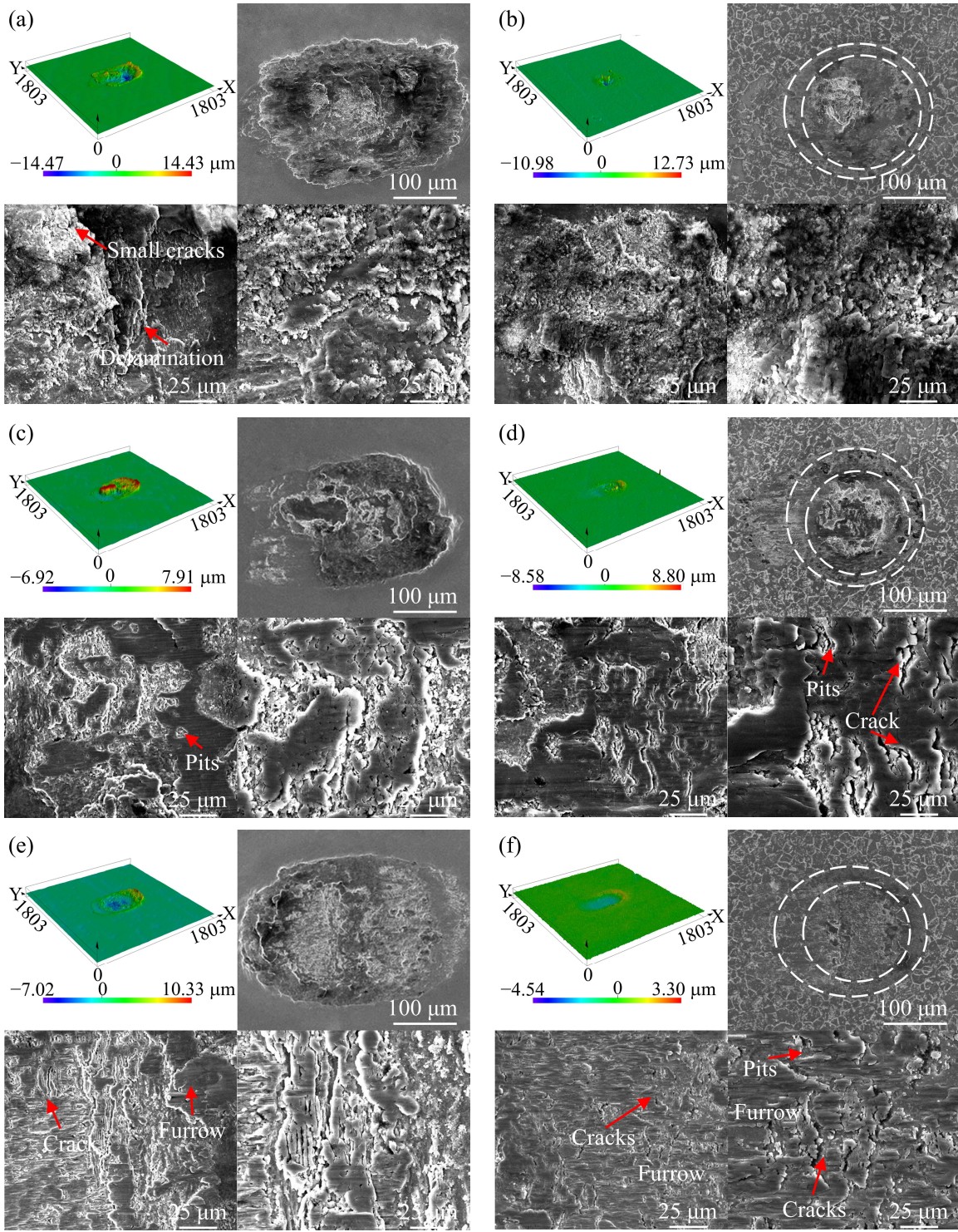

**Figure 3.** SEM images of wear marks of AISI 316L steel at (**a**) 30 N, (**c**) 50 N, and (**e**) 70 N. SEM images of wear marks of the carburized layer at (**b**) 30N, (**d**) 50N, and (**f**) 70N.

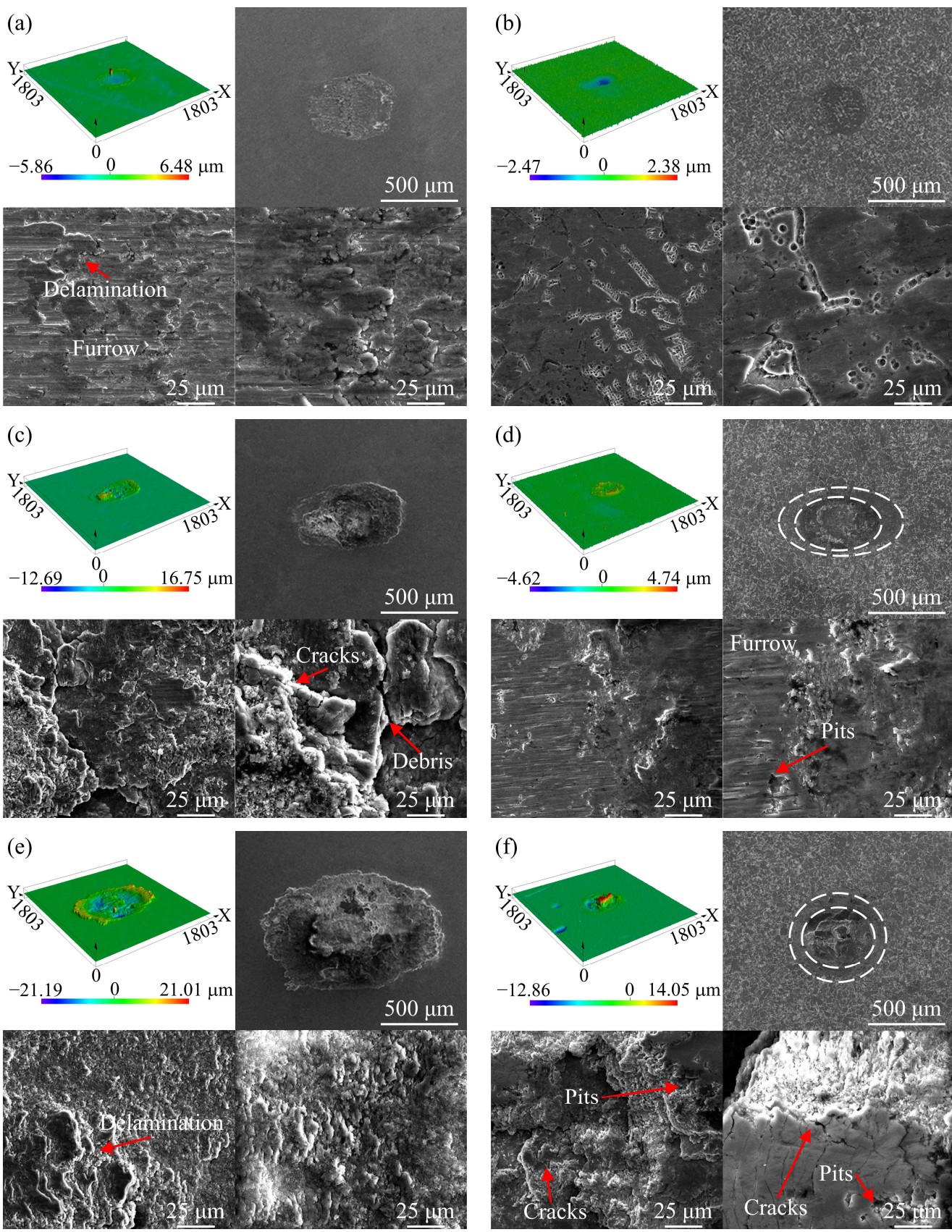

**Figure 4.** SEM images of wear marks of AISI 316L steel under displacement of (**a**) 50 μm, (**c**) 75 μm, and (**e**) 100 μm. SEM images of wear marks of the carburized layer under displacement of (**b**) 50 μm, (**d**) 75 μm, and (**f**) 100 μm.

The distributions of oxygen elements in the wear marks of the AISI 316L steel and carburized layer under varying load and displacement conditions are shown in Figure 5. Both the untreated AISI 316L steel and carburized layer showed significant oxygen accumulation in the wear marks under varying load and displacement conditions. Thus, fretting wear involves oxidizing wear.

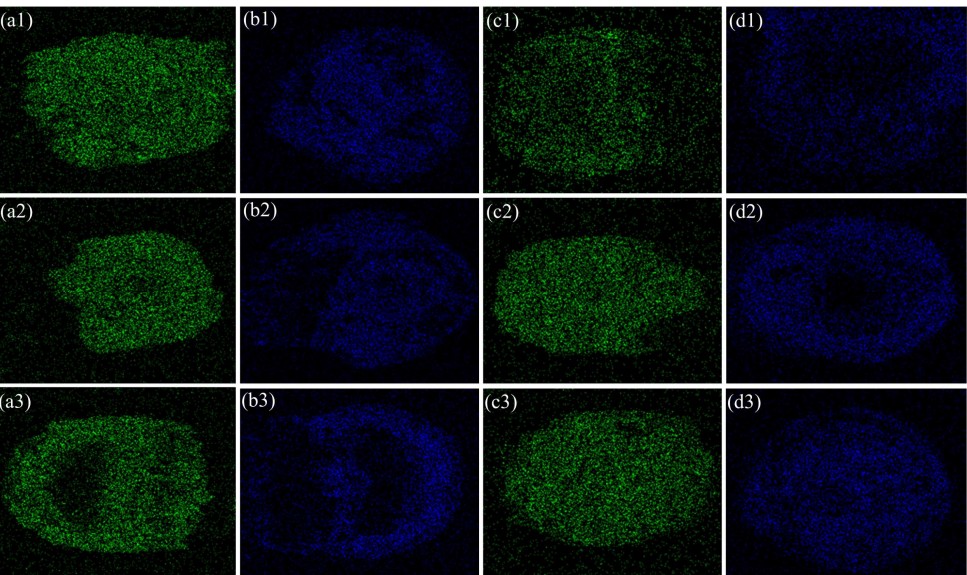

**Figure 5.** Oxygen element distribution of AISI 316L steel under loads of (**a1**) 30 N, (**a2**) 50 N, and (**a3**) 70 N and displacements of (**c1**) 50 μm, (**c2**) 75 μm, and (**c3**) 100 μm. Oxygen element distribution of the carburized layer under loads of (**b1**) 30 N, (**b2**) 50 N, and (**b3**) 70 N and displacements of (**d1**) 50 μm, (**d2**) 75 μm, and (**d3**) 100 μm.

Under varying load and displacement conditions, the morphology of the wear marks is determined by the competitive influence of mechanical and thermochemical factors. When mechanical factors are dominant, fretting wear mainly involves plastic deformation, adhesive wear, and fatigue. This leads to the formation of contact interface crystals with high-density dislocations [37]. When thermochemical factors are dominant, the contact interface generates heat that reaches "friction flashover temperature" during fretting wear, leading to the occurrence of many heat-promoted interfacial chemical reactions. The micro-convex body in close contact with the surface of the friction pairs is oxidized to form an oxide film, which reduces the direct contact of friction pairs. Consequently, the friction coefficient is reduced. As this oxide film is continually formed and destroyed in the middle and late stages of wear, wear chips are formed at the contact interface, which aggravates abrasive wear and oxidation wear. Thus, the friction coefficient is enhanced.

The abrasion morphology of the GCr15 ball under variable loads and displacements is shown in Figure 6. Under different load and displacement conditions, the wear surface area of the upper sample ball in GCr15/PC was smaller than that of the upper sample ball in GCr15/316L. Thus, the wear degree of the upper sample ball was reduced due to plasma carburization. With increasing load and displacement, the center of the GCr15 ball wear marks was depressed due to material spalling, and some central areas showed circular bumps formed by debris accumulation. The edges of these wear marks showed slight plastic upheaval, wear debris accumulation, and plastic deformation. The wear at the center of the wear marks was more significant than that at the edge. This is potentially because the wear chips generated in the central area cannot be easily discharged during the ball–plane fretting wear process. Thus, these wear chips accumulate, resulting in a varying load distribution on the contact surface. The large pressure applied to the center of the wear chips leads to the destruction of the central area first. The accumulated debris

particles also easily aggravate fretting wear, and the wear spreads from the center to the edge of the wear chips.

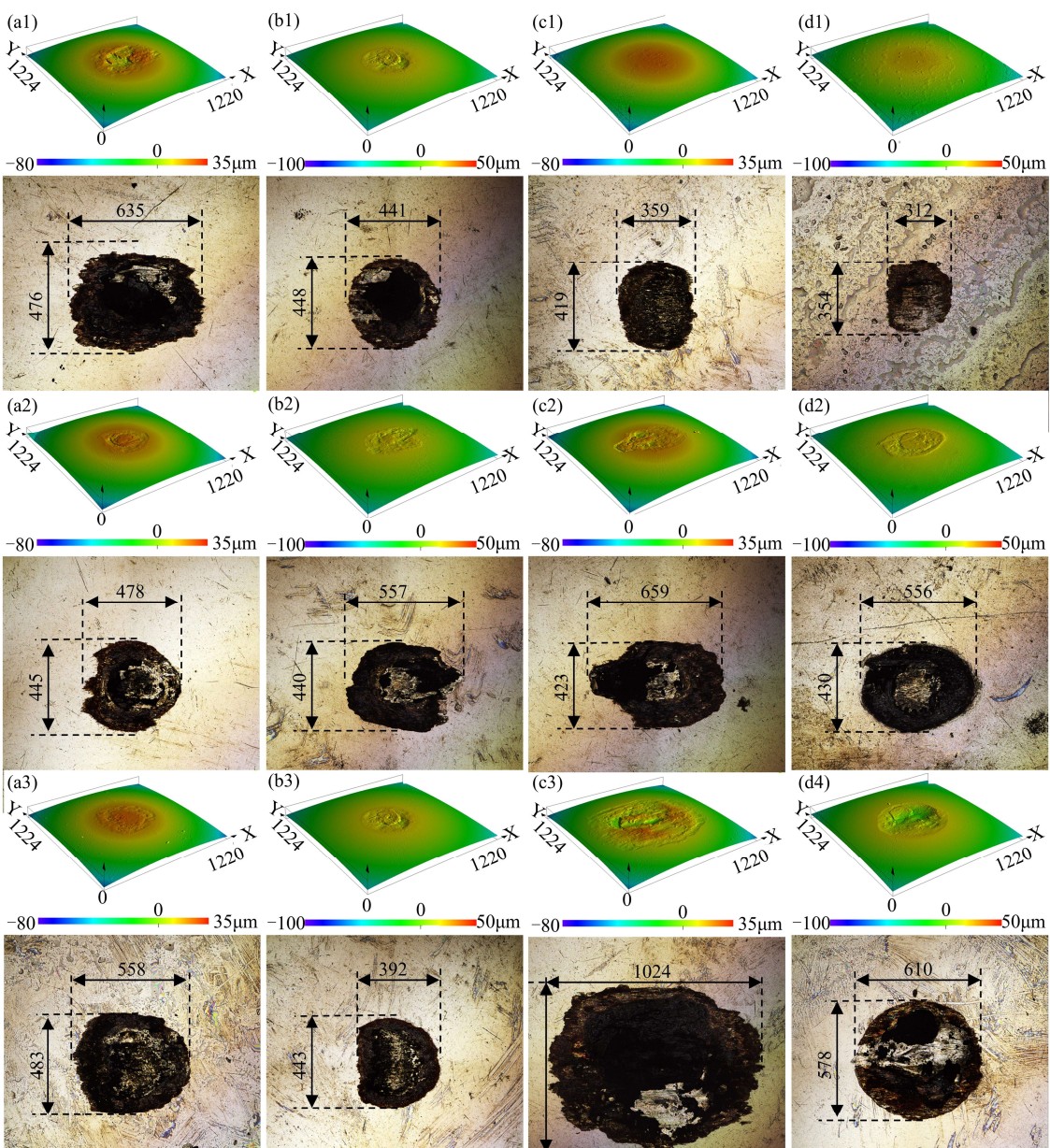

**Figure 6.** OM images of wear marks of the GCr15 ball under loads of (**a1**) 30 N, (**a2**) 50 N, and (**a3**) 70 N and displacements of (**c1**) 50 μm, (**c2**) 75 μm, and (**c3**) 100 μm. OM images of wear marks of the GCr15 ball under loads of (**b1**) 30 N, (**b2**) 50 N, and (**b3**) 70 N and displacements of (**d1**) 50 μm, (**d2**) 75 μm, and (**d3**) 100 μm.

### 3.4. Wear Profile Analysis

The X-axis contour in the same direction as the wear and movement was selected for characterization. The wear profiles of the AISI 316L steel and carburized layer under different loads and displacements are shown in Figure 7. These wear profiles show wave fluctuations, which occur because the GCr15 ball and AISI 316L steel are not ideal planes. Thus, when the GCr15 ball and AISI 316L steel surfaces come into contact and fretting wear occurs, higher micro-convex bodies come into contact first and are worn out, while lower micro-convex bodies do not come into contact yet and are protected, resulting in uneven wear [38]. During the reciprocating movement of the GCr15 ball, some of the steel

material is pushed to the front end of the ball under the action of cutting and rolling. The dislocation of internal material forms a micro-work hardening zone. When the GCr15 ball passes through this zone, slight wearing occurs in some areas and heavy wearing in other areas. This is a cyclic process, leading to fluctuations in the wear profile. In this study, under variable load and displacement conditions, the wear depth of GCr15/PC was lower than that of GCr15/316L. The carburized layer had a wear depth of up to 8 μm, and this layer was not completely worn through. Thus, the carburized layer provided better protection than the AISI 316L steel surface. The three-dimensional topography of the wear marks shows that the wear contour was higher than the base level. This indicates the accumulation of wear chips inside the wear marks of the AISI 316L steel and carburized layer. Thus, the adhesive wear and plastic deformation were serious.

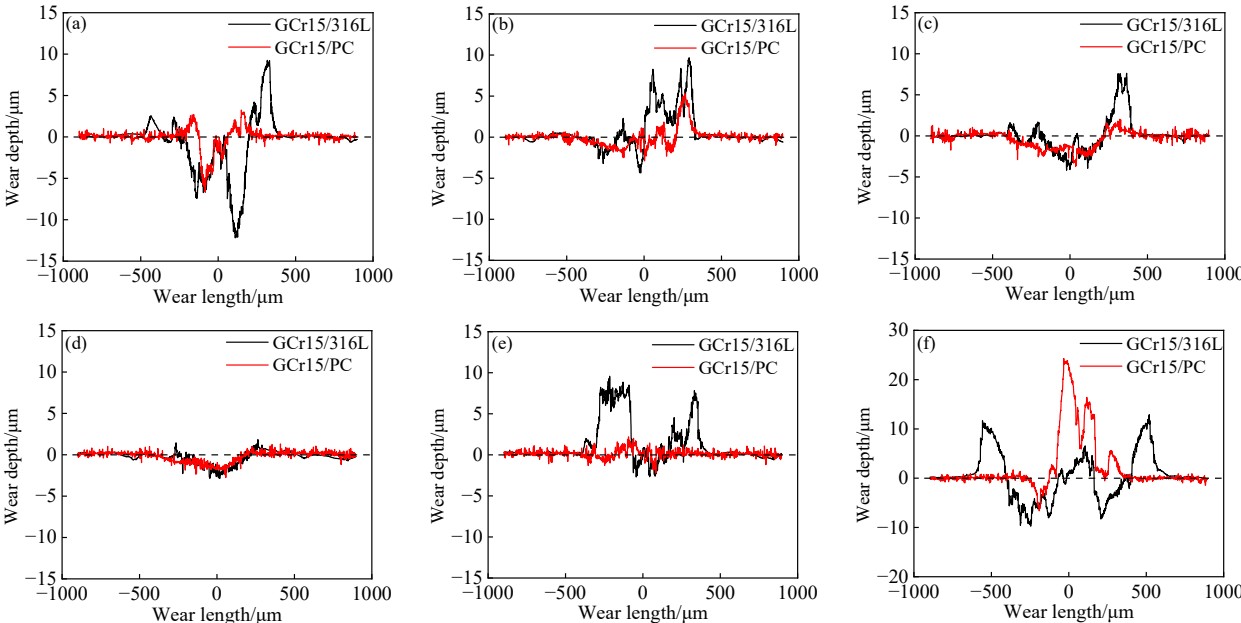

**Figure 7.** Wear profiles of the AISI 316L steel and carburized layer under loads of (**a**) 30 N, (**b**) 50 N, and (**c**) 70 N and displacements of (**d**) 50 μm, (**e**) 75 μm, and (**f**) 100 μm.

Under variable load conditions, the wear profile of GCr15/316L changed from W-shaped to V-shaped. In contrast, the wear profile of GCr15/PC changed to a W-V-U-type profile. Under variable displacement conditions, the wear profile of GCr15/316L changed to a V-M-W-type profile, while that of GCr15/PC changed to a V-W-M-type profile.

When bidirectional adhesive wear occurred in the friction pairs, the wear width was significantly smaller than the wear depth, and the wear profile showed a V shape. Point contact was observed between the upper and lower samples, and the wear marks tended to rapidly expand in the depth direction. During the wear process, the GCr15 ball continuously ploughed in the transverse and depth directions, the material fell off in a layer-by-layer manner, the width of the base continuously expanded to form a U shape, the point contact of the upper and lower samples changed to approximate surface contact, and the bottom of the wear mark became smoother. When the center of the abrasion exhibited a more significant degree of adhesive wear than the edge, material tearing and spalling occurred at the edge before the center. Consequently, a W-shaped wear profile was obtained. Under the heat generated by friction and the reciprocating motion of the upper sample, the W-shaped wear profile evolved into an M-shaped profile when significant debris accumulation occurred at the edge.

### 3.5. Wear Rate and Wear Volume Analysis

The wear volume and wear rate were used to evaluate the wear resistance of the carburized layer. The wear rate $K$ (mm$^3$·(N·m)$^{-1}$) can be calculated as follows [39]:

$$K = \frac{V}{4NDF} \tag{1}$$

where $V$ is the wear volume (mm$^3$), $N$ is the number of cycles, $D$ is the displacement (m), and $F$ is the normal load ($N$).

The wear volumes and wear rates of GCr15/316L and GCr15/PC under variable load and displacement conditions are displayed in Figure 8. As shown in Figure 8a, the wear volume and wear rate of GCr15/316L and GCr15/PC decreased with increasing load. Several factors influenced this trend. (1) With increasing load, the frictional heat and stress at the wear interface increased and continuously accumulated. This provided the contact surface with a higher activation energy. The wear surface underwent a frictional glazing effect under the combined action of heat and chemical reactions, leading to the formation of a glazing film. This film weakened the wear. A greater load leads to more significant glazing on the wear surface and a thicker glazing film [40,41]. (2) Work hardening occurred on the wear surface of the matrix under severe mechanical catalysis, and the concentration of residual stress at the surface increased the hardness and mechanical strength of the contact surface. Thus, fretting wear damage to the matrix was reduced. (3) When the normal load increased, some of the third body layer was further refined and transformed into amorphous products (which are highly brittle) and spalled to form abrasive chips during wear. The formation of these abrasive chips accelerated the oxidation of the wear interface, and the chips played a lubricating role [42–44]. Thus, the wear rate and wear volume decreased with increasing normal load. GCr15/316L and GCr15/PC exhibited their lowest wear volumes and wear rates under a load of 70 N.

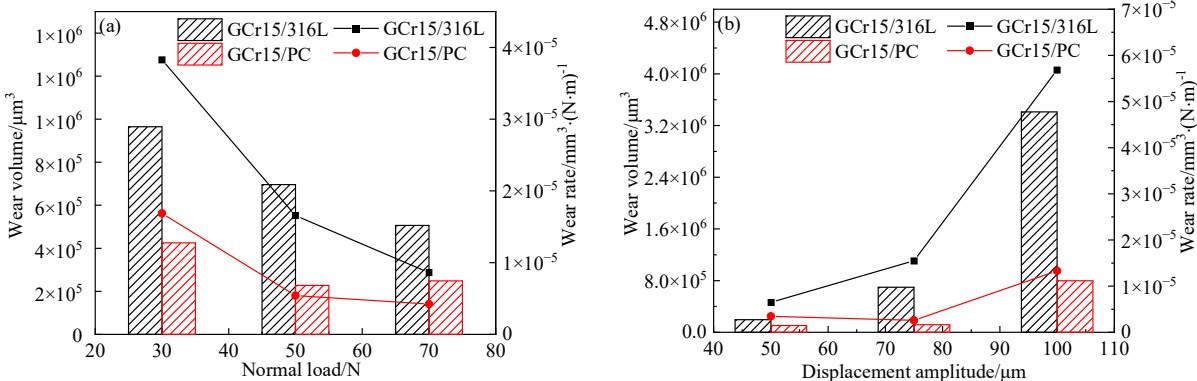

**Figure 8.** Wear volume and wear rate of the untreated and carburized AISI 316L steel under different (**a**) loads and (**b**) displacements.

The wear volume and wear rate of GCr15/316L and GCr15/PC increased with increasing displacement, as shown in Figure 8b. As the actual contact area of the upper and lower samples increased with increasing displacement, the transverse wear was enhanced and there was serious material damage. In contrast, under small displacements, more abrasive particles remained in the contact area to form the third body layer, which reduced direct contact and friction between the upper and lower samples. Thus, the load distribution on the contact interface was more uniform under small displacements. This led to a low wear rate and wear volume. With increasing displacement, the abrasive particles were more easily discharged from the contact area, resulting in direct contact between friction pairs and serious wear damage. GCr15/316L and GCr15/PC exhibited their highest wear volumes and wear rates under a displacement of 100 μm.

The wear volume and wear rate of GCr15/PC under variable load and displacement conditions were significantly lower than those of GCr15/316L. As shown in Figure 7, the maximum wear depth of the carburized layer was only 8 µm and it was not completely worn through. This indicates that the carburized layer has good wear resistance and protects the AISI 316L steel. The single Sc phase of the carburized layer reduced plastic deformation during fretting wear, and the surface of this layer did not easily form adhesion points. Moreover, the carburized layer had a uniform and dense structure, and there were no obvious defects such as holes or cracks. This improved the adhesion resistance of the surface and the ability of the carburized layer to resist elastoplastic deformation. Therefore, the presence of the carburized layer leads to improved anti-fretting wear performance and reduces wear, spalling, and cutting on the AISI 316L steel surface.

## 4. Discussion

### 4.1. Fretting Wear Process Analysis

The wear process of GCr15/PC and GCr15/316L can be divided into three stages, as shown in Figure 9. (1) Initial stage I: The GCr15 ball forms a micro-convex contact point with the carburized layer and AISI 316L steel. The contact mode is sphere–micro-convex contact, and the maximum contact stress is experienced at the initial moment of wear. (2) Wear stage II: As wear continues, the micro-convex body on the surface of the upper and lower samples is worn out, and the wear contact interface begins to produce wear chips. Acting as the third body, these wear chips gradually participate in the wear process, resulting in the formation of damage morphologies such as adhesion, cracks, scratches, and small wear pits. At this time, three-body contact exists between the GCr15 ball wear micro-plane, the carburized layer (or AISI 316L steel) concave surface, and the wear chips. The roughness of the micro-convex body is reduced, contact stress is reduced, and the wear chips are filled and compacted to form a cabochon–plane contact. (3) Stability stage III: In the middle–late stages of wear, the contact surface formed after the GCr15 ball wear process tends to be flat or concave, and the micro-convex body completely disappears. Thus, a stable chip layer (the third body layer) is formed. The state of the wear interface remains relatively stable, contact stress remains constant, and the contact mode is plane–plane contact or concave plane–plane contact. In this stage, damage morphologies such as severe adhesion, furrows, fatigue peeling, and large wear craters are formed.

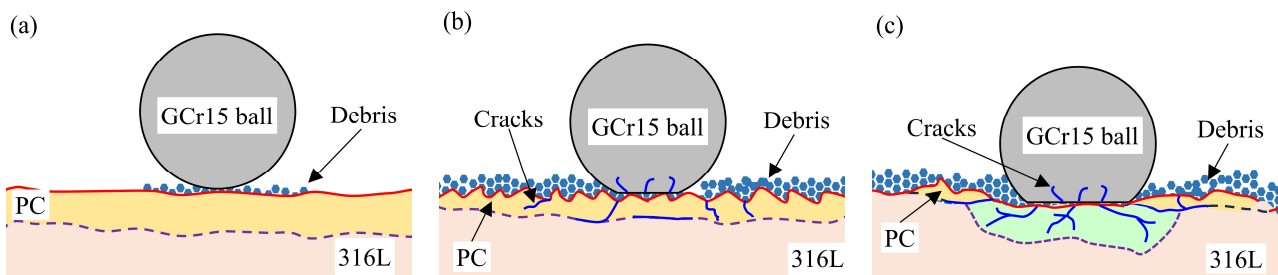

**Figure 9.** Fretting wear process and mechanism of GCr15/PC and GCr15/316L at (**a**) initial stage, (**b**) wear stage, and (**c**) stability stage.

### 4.2. Cutting Plasticity Ratio Analysis

The cutting plastic ratio ($f_{cp}$) was used to quantitatively analyze the proportion of micro-cutting and micro-ploughing in material deformation. To calculate $f_{cp}$, the wear contours were simplified, as shown in Figure 10a, according to [45]. The formula used to calculate $f_{cp}$ is as follows:

$$f_{cp} = \frac{A_2 - A_1}{A_2} \tag{2}$$

where $A_1$ is the cross-sectional area of the material build-up to the edge of the wear mark (mm²) and $A_2$ is the cross-sectional area of the wear mark (mm²).

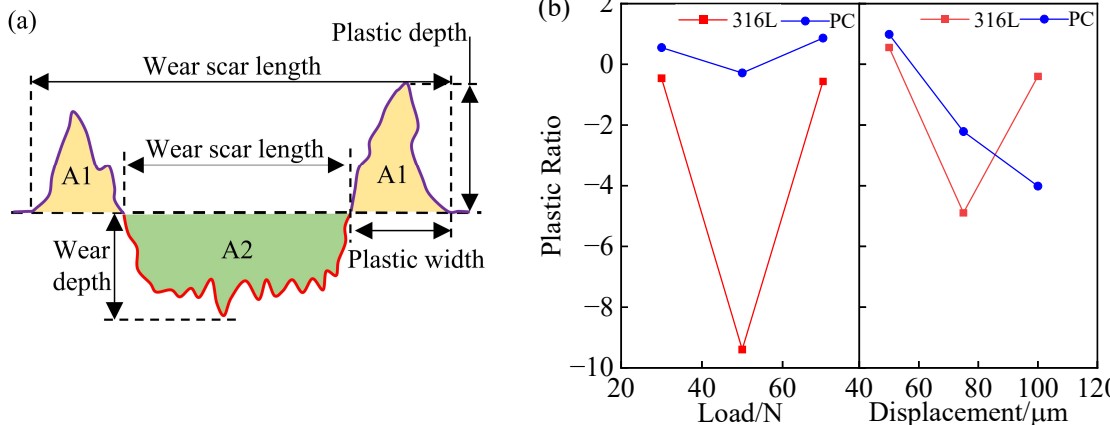

**Figure 10.** (**a**) Simplified wear contours and (**b**) cutting plasticity ratios of AISI 316L steel and carburized layer.

As shown in Figure 10b, the $f_{cp}$ values of GCr15/316L under variable load conditions were all less than 0, indicating that the adhesion effect had a dominant influence on material damage, while micro-cutting (ploughing) only had a slight effect. The $f_{cp}$ values of GCr15/PC were greater than 0, indicating that the micro-cutting (ploughing) effect was more significant. Thus, the carburized layer reduced the degree of adhesive wear in the AISI 316L steel under varying load conditions. Under variable displacement conditions, the $f_{cp}$ values of GCr15/316L and GCr15/PC were greater than 0 when the displacement was 50 μm and less than 0 under other displacements. This demonstrates that the adhesion effect had a dominant influence on material damage under variable displacement conditions. The load influenced the wear mechanism of both untreated AISI 316L steel and the carburized layer more significantly than displacement. In combination with the local SEM images of the wear marks shown in Figures 3 and 4, it can be seen that the material can be cut off via micro-cutting to form wear chips. Through the interaction of ball migration, tile laying, compaction, and high-temperature sintering, the morphology of the wear marks shows one or several kinds of adhesion, wear particles, and a fatigue layer. Micro-ploughing forms furrow or groove morphologies, and the wear chips are deposited at the edge of the wear mark together with the sample via plastic upheaval. The adhesion effect leads to adhesion wear and plastic upheaval.

*4.3. Frictional Dissipation Energy Analysis*

The frictional dissipation energy coefficient was used to evaluate the stability of the AISI 316L steel fretting wear process before and after plasma carburization. The frictional dissipation energy $E_d$ (J) can be represented as

$$E_d = \sum_{i=1}^{i=N} 4\mu F_i D_i \tag{3}$$

The frictional dissipation energy coefficient $\alpha_e$ (mm$^2 \cdot$J$^{-1}$) can be calculated as follows [46]:

$$\alpha_e = \frac{V}{E_d} \tag{4}$$

where $V$ is the wear volume (mm$^3$), $N$ is the number of cycles, $D_i$ is the displacement (m), $F_i$ is the normal load (N), and $\mu$ is the friction coefficient.

The frictional dissipative energy and friction dissipation energy coefficients of GCr15/316L and GCr15/PC under different working conditions are shown in Figure 11. The frictional dissipation energy coefficients of GCr15/PC under varying load and displacement conditions were lower than those of GCr15/316L, indicating that LTPC can improve the stability of the AISI 316L steel fretting wear process.

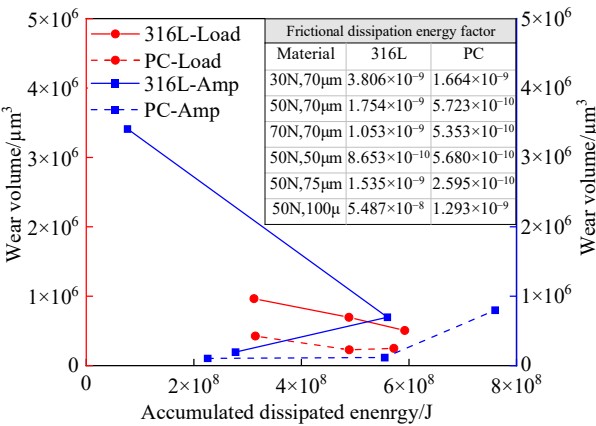

**Figure 11.** Frictional dissipative energy and friction dissipation energy coefficients of the untreated and carburized AISI 316L steel under different loads and displacements.

With increasing load, the frictional dissipation energy coefficients of GCr15/316L and GCr15/PC decreased, and the fretting wear processes of GCr15/316L and GCr15/PC gradually became stable. Under a small load, large hard-phase abrasive particles were generated at the wear interface, and the roughness of the contact interface of the friction pair was relatively high. This means that the friction pairs had high frictional dissipation energy. With increasing load, the wear chips were refined and quickly discharged from the contact area, and two-body wear gradually transformed into three-body wear. A dynamic balance was achieved between the generation and discharge of wear chips, the contact interface became relatively smooth, and the number of micro-convex bodies decreased. Material loss at the contact interface mainly occurred via shearing. Due to the plastic stability of the material at this point, the wear interface tended to be stable. Thus, the frictional dissipation energy coefficient showed a downward trend.

Under variable displacement conditions, the frictional dissipation energy coefficient of GCr15/316L continually increased, while the frictional dissipation energy coefficient of GCr15/PC first decreased and then increased. Both GCr15/316L and GCr15/PC showed unstable fretting wear processes. Due to the low hardness of the AISI 316L steel, adhesion easily occurred during wear. With increasing displacement, the actual contact area of the upper and lower samples increased, transverse wear intensified, the material damage became more serious, peeling and micro-cracks appeared, the state of the wear interface became unstable, and the frictional dissipation energy of GCr15/316L increased. In contrast, the high hardness and toughness of the carburized layer meant that GCr15/PC only exhibited slight interfacial fretting damage under small displacements. The elastic deformation of the friction pairs dominated the relative motion of fretting wear, leading to a low frictional dissipation energy coefficient. When displacement increased, local material spalling and micro-cracks appeared in the wear marks. Wear and local fatigue were competitive and dynamic processes at this time. During the fretting wear process, some energy was absorbed by the initiation and expansion of micro-cracks, and the frictional dissipation energy coefficient decreased. Under a large displacement, the fretting damage of GCr15/PC was further aggravated, and the presence of a large number of hard-phase wear chips on the contact surface led to the violent and disorderly vibration of the wear interface during fretting wear. This caused high levels of frictional energy dissipation during fretting wear, and the frictional dissipation energy coefficient increased [7].

## 5. Conclusions

In this work, the effect of double-glow low-temperature plasma carburization on the fretting wear behavior of 316L austenitic stainless steel under varying load and displacement conditions was investigated. The untreated and carburized samples were examined

with SRV-V, SEM, XRD, and 3D confocal laser microscopy. The results are summarized as follows:

1. The carburized layer was composed of a single Sc phase, which exhibited good uniformity and continuity. This layer was metallurgically combined with the matrix. Plasma carburization increased the surface hardness of the AISI 316L steel by a factor of approximately four.

2. Under varying load conditions, the wear mechanism of GCr15/316L changed from adhesive wear and abrasive wear to adhesive wear, fatigue peeling, and abrasive wear. The wear mechanism of GCr15/PC changed from adhesive wear to adhesive wear and fatigue delamination, accompanied by a furrowing effect. Under variable displacement conditions, both GCr15/316L and GCr15/PC mainly exhibited adhesive wear and fatigue peeling. Oxygen accumulated in the wear marks of both the AISI 316L steel and the carburized layer, indicating oxidative wear.

3. At higher loads and displacements, the frictional dissipation energy coefficient and wear rate of GCr15/PC were lower than those of GCr15/316L. Moreover, the carburized layer showed better fretting wear resistance. Plasma carburization improved the stability of the AISI 316L steel fretting wear process and changed the fretting regime of the AISI 316L steel.

4. The wear depth of GCr15/PC under variable load and displacement conditions was lower than that of GCr15/316L, showing that the carburized layer can effectively protect AISI 316L steel. Under variable load conditions, the wear profile of GCr15/316L changed from W-shaped to V-shaped, while that of GCr15/PC changed to a W-V-U profile. Under variable displacement conditions, the wear profile of GCr15/316L changed to a V-M-W profile, while that of GCr15/PC changed to a V-W-M profile.

5. This study did not provide a more in-depth discussion on the existence of interfacial abrasive debris and its influence on fretting wear behavior and did not analyze the fretting wear mechanism of the subsurface under variable loads and displacements. This study has shown that the carburized layer with high surface hardness, as well as superior resistance to fretting wear, along with a reduction in wear rate and frictional dissipation energy coefficient, can all be considered as anti-wearing coatings of ball valves.

**Author Contributions:** L.S. prepared the samples, performed the experiments, and wrote the manuscript. Y.L. edited and reviewed the manuscript. C.C. prepared the carburized samples. G.B. and X.L. contributed to the technical discussions of the results. All authors have read and agreed to the published version of the manuscript.

**Funding:** This research was funded by the Key Science and Technology Project of Gansu Province, China (No. 20YF8GA058); Science and Technology Project of Wenzhou City, China (No. ZG20211003); and Major Science and Technology Project of Gansu Province, China (No. 22ZD6GA008).

**Institutional Review Board Statement:** Not applicable.

**Informed Consent Statement:** Not applicable.

**Data Availability Statement:** Data are contained within the article.

**Conflicts of Interest:** The authors declare no conflicts of interest.

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
