# Peer review of "Effect of Low-Temperature Plasma Carburization on Fretting Wear Behavior of AISI 316L Stainless Steel"

_coatings, doi:10.3390/coatings14020158_

Round 1
Reviewer 1 Report
Comments and Suggestions for Authors
The research is on Effect of low-temperature plasma carburization on fretting 2 wear behavior of AISI 316L stainless steel. The work can be considered after addressing following comments-
The abstract should be modified by highlighting important results in terms of % increase/decrease value. The potential application is also duly endorsed at the end of the abstract.
The literature section is poorly written as the papers considered are not recent. So, adding a minimum of 5 to 6 papers from 2023 is advised.
The rationale for selection of material is not mentioned at the end of the abstract.
The novelty and clear objective should be mentioned at the end of abstract.
Restructure the headings into standard journal format i.e 1. Introduction, 2. Materials and Methods, 3. Results and discussion, 4. Conclusion and References.
Line 186: “The friction coefficient curves of GCr15/316L and GCr15/PC under different loads and displacements are shown in Figure 2. These curves are divided into three stages”- It is advised to add the following articles for literature proof.
https://www.sciencedirect.com/science/article/pii/S0043164819302005
https://www.scielo.br/j/mr/a/ZNjNwW3VMVSh3n83KzjF7ww/?lang=en
Line 188: In Stage I description: “The friction coefficient curves of GCr15/316L and GCr15/PC sharply rise and very rapidly reach a maximum. In the initial running-in stage, the friction pair experiences slightly convex contact, the relative contact area is small, and fewer hard phase particles are shed, resulting in a low friction coefficient.” The reasoning should be explained through a literature proof.
Similarly, add literature proof for the reasoning in Stage II and III's description.
At the end of the conclusion, include the drawbacks of the current study and future scope.
Comments on the Quality of English LanguageThe research is on Effect of low-temperature plasma carburization on fretting 2 wear behavior of AISI 316L stainless steel. The work can be considered after addressing following comments-
The abstract should be modified by highlighting important results in terms of % increase/decrease value. The potential application is also duly endorsed at the end of the abstract.
The literature section is poorly written as the papers considered are not recent. So, adding a minimum of 5 to 6 papers from 2023 is advised.
The rationale for selection of material is not mentioned at the end of the abstract.
The novelty and clear objective should be mentioned at the end of abstract.
Restructure the headings into standard journal format i.e 1. Introduction, 2. Materials and Methods, 3. Results and discussion, 4. Conclusion and References.
Line 186: “The friction coefficient curves of GCr15/316L and GCr15/PC under different loads and displacements are shown in Figure 2. These curves are divided into three stages”- It is advised to add the following articles for literature proof.
https://www.sciencedirect.com/science/article/pii/S0043164819302005
https://www.scielo.br/j/mr/a/ZNjNwW3VMVSh3n83KzjF7ww/?lang=en
Line 188: In Stage I description: “The friction coefficient curves of GCr15/316L and GCr15/PC sharply rise and very rapidly reach a maximum. In the initial running-in stage, the friction pair experiences slightly convex contact, the relative contact area is small, and fewer hard phase particles are shed, resulting in a low friction coefficient.” The reasoning should be explained through a literature proof.
Similarly, add literature proof for the reasoning in Stage II and III's description.
At the end of the conclusion, include the drawbacks of the current study and future scope.
Author Response
1.Based on the reviewers' comments, we have made appropriate revisions to the abstract. Some old references in the abstract have been replaced. The structure of the article has been adjusted. The modifications are also marked in red.
2.Based on the reviewers' comments, we have added the corresponding references for literature proof. The modifications are also marked in red.
3.Based on the reviewers' comments, we have added the shortcomings and future developments of this study at the end of the conclusion. The modifications are also marked in red.

Reviewer 2 Report
Comments and Suggestions for Authors
The authors have to answer the following questions.
1. Fig. 1(a) needs proper distinction between the substrate and the carburized layer. The hardness value requires the error value. The experiment about the hardness measurement has not mentioned. The line no. 143-145 requires proper references.
2. Good quality (high magnification) SEM microstructure of the carburized layer is missing in the manuscript.
3. It is difficult to understand that from where Sc is coming to the carburized layer. The microstructure requires further EDX mapping and composition. Moreover, how the Sc-containing phase effects the friction behavior of the layer.
4. The authors need to explain the surface roughness with the friction coefficient of the layer. The loads used for measurement of the hardness and the friction tests are required. Refer the article by Sarangi et al. (Pulse electrodeposition and characterization of graphene oxide particle-reinforced Ni–W alloy matrix nanocomposite coatings) to explain this correlation.
5. The heading of each section should start from 1 not 0. Section 2 (should be 3 in the revised version) has mentioned "Results and discussion", because there is a separate "Discussion" is there.
Comments on the Quality of English Language
English writing skill should be improved.
Author Response
1.Based on the reviewers' comments, we have made a proper distinction between the matrix and the carburized layer. We have added the parameters of the hardness measurement and the error values in “2.3. Performance testing and characterization”. We have added the corresponding references to line no.143-145 for literature proof . The modifications are also marked in red.
The modified contents are as follows.
The hardness measurements were carried out using a 20 gf load applied for a duration of 15 s. The surface hardness of the carburized layer was 897±18 HV0.2, which is approximately four times higher than that of the untreated AISI 316L steel surface (273±33 HV0.2).
2.Based on the reviewers' comments, the high magnification SEM microstructure of the carburized layer has been added.
3.Based on the reviewers' comments, we have added the metallographic photographs and EDS pattern of the carburized layer.
Similar to low-temperature plasma nitriding technology, low-temperature plasma carburization technology is a sputtering-absorption-desorption-diffusion process. We use the schematic diagram of the formation of the nitriding layer (see Fig 1) to illustrate it[1]. In the glow ion diffusion furnace, under vacuum, the samples are used as the cathode, and a pulsed high-voltage current is applied to produce a glow discharge to ionize the carbon-containing medium (the carburizing gases are C2H2 and H2). The ionized carbon ions bombard the surface of the samples and penetrate into the surface while the samples are heated up, then continue to diffuse into the matrix(see figure 2). The carbon atoms that diffuse into the matrix will combine with the iron atoms inside the matrix in solid solution to form Sc phase. It can lead to the expansion and distortion of the matrix lattice structure. This process can obtain the carburized layer of a certain thickness, which contain the Sc phase mainly.
The presence of the Sc phase increases the surface hardness and residual stress. It can also improve the surface roughness of AISI 316L steel slightly. In addition, the Sc phase causes the lattice structure of the AISI 316L steel to undergo expansion and distortion which affects the fretting wear behavior in terms of surface hardness, roughness and crystal structure. The carburized layer can improve the stability of the AISI 316L steel fretting wear process, decrease the wear rate, changed the fretting regime , and protect the AISI 316L steel.
References:
[1] WANG Zilong. The Study on Low Temperature Plasma Nitriding of Several Stainless Steels [D].Lanzhou University of Technology, 2016.
[2] Sarangi C K, Sahu B P, Mishra B K, et al. Pulse electrodeposition and characterization of graphene oxide particle-reinforced Ni–W alloy matrix nanocomposite coatings[J].Journal of Applied Electrochemistry, 2020, 50(2):265-279.
[3] A Z H D, B W Z, B H W K, et al. Surface hardening of laser melting deposited 12CrNi2 alloy steel by enhanced plasma carburizing via hollow cathode discharge[J].Surface and Coatings Technology, 2020, 397.
[4] Hu, G.; Cai, X.; Rong, Y. Material Science Foundation[M]. Shanghai: SHANGHAI JIAO TONG UNIVERSITY PRESS, 2010.

Reviewer 3 Report
Comments and Suggestions for Authors
Title: Effect of low-temperature plasma carburization on fretting wear behavior of AISI 316L stainless steel
Overview and general recommendation:
In this study the authors investigate the effect of low-temperature plasma carburization on the wear resistance of 316L stainless steel. I found the paper to be overall well written and much of it to be well described. However, I have a few concerns about the overall fit of the work with the current special issue, I am listing possible points that should be additionally discussed in the manuscript before considering for publication; therefore, I recommend that a major revision is warranted. I explain my concerns in more detail below.
Major Comments:
1. From the introduction, it is unclear what is the novelty of this work is and how it fits with the scope of the current special issue. For example, is “coating” of the surface via LTPC new, or is the formation of a metal layer on the surface via LTPC new, or is unveiling the mechanism for the wear resistance new, or is the “characterization” method for the wear experiments new?
2. Slightly relating to the first comment, it feels as there should be more characterization of the carburized layer if it is to be suited for publication in “Coatings”. For example, the authors claim that the layer is “evenly distributed, continuous, dense, and metallurgically combined” in lines 135-136. This seems premature to simply conclude visually from just the cross-sectional SEM image alone.
3. In lines 143-145 the authors mention that the carbon atoms are soldered in the gaps of the austenite structure. How is this proven? Moreover, if there are a significant amount of carbon species in the layer, shouldn’t there be a weak but broad peak in the XRD spectra?
4. In addition to the comment 3, it is expected that there is residual internal stress built up if the carbon atoms are lodged in the gaps. If this is the case for the treated surface, I would expect that the slight deformation of the layer via fretting would induce a rapid release of the stress in the form a huge crack generation. Rather it seems that crack generation is less for the treated surface. Any comments on this?
5. In lines 141-142 the authors mention that the carburized surface has a slightly higher surface roughness than untreated steel surface due to the crystal structure. Why does the overall frictional coefficient become significantly lower than that of the untreated surface in stage II? If the roughness is due to crystal structure, digging into the material and forming an oxide layer seems to be insufficient in explaining this large change? Any thoughts?
Minor Comments:
1. Minimal abbreviations in the abstract are desired. Difficult to get a general idea of the study without reading through the main text or having significant background knowledge.
2. Captions for Figure 5 and beyond should not abbreviate the experimental parameters. Very difficult to understand which figure corresponds to what type of experimental situation without going back to the previous figures.
3. Figure 8: Cannot tell if the bar or line is corresponding to wear volume or wear rate.
Comments on the Quality of English Language
Minor edits are recommended to make the sentences easier to read. Some are too long or redundant.
Author Response
1.(1) Currently, studies investigating the wear behavior of austenitic stainless steel have mainly focused on evaluating the conventional reciprocating friction and wear behavior. However, little attention has been paid to the fretting wear behavior and mechanism of austenitic stainless steel. Fretting wear are covert, catastrophic, and universal, known as "industrial cancer". It can cause friction and wear of components, the formation of pollution sources, and accelerate fatigue crack initiation and propagation. Fretting wear can significantly reduce the service life of components[1]. (2) AISI 316L steel exhibits various disadvantages, including low hardness, poor wear resistance, and easy adhesion to abrasive parts. Moreover, the performance of AISI 316L steel cannot be improved through traditional phase transformation. These draw-backs significantly reduce the service life of AISI 316L stainless steel parts and affect their application range[2-4].(3) Hard sealing fixed ball valve stem/bearing, bearing/body, and spool/sealing surfaces are prone to fretting wear when the valve is normally open or normally closed. Fretting wears the contact interface between valve parts and accelerates the further initiation and expansion of the original cracks on the material surface. This results in the local failure of the bearings and spool, as well as the leakage failure of the sealing surface[1].
Based on the above three scientific issues, this paper selects AISI 316L stainless steel as the lower specimen(the ball valve commonly used as a typical material) and GCr15 ball as the upper specimen. To improve the hardness and wear resistance of stainless steel, we use the low-temperature plasma carburization (LTPC) technology to prepare a carburized layer on the surface of stainless steel. The influence of LTPC on the fretting wear behavior of AISI 316L steel under varying loads and displacements is investigated. Our research methods extend from dry sliding frictional wear to fretting wear, broadening the range of applications for carburized layer. Therefore, the coating preparation process, revealing wear resistance mechanism and wear test characterization methods used in this paper are new.
References:
[1] ZHU, M, H, CAI, Z B, Zhou, Z R. Fretting Wear Theory[M]. Beijing: Science Press, 2021.
[2] Godec M , Donik C , Kocijan A ,et al.Effect of post-treated low-temperature plasma nitriding on the wear and corrosion resistance of 316L stainless steel manufactured by laser powder-bed fusion[J].Additive Manufacturing, 2020(32-):32.
[3] Peng Y , Liu Z , Chen C ,et al.Effect of low-temperature surface hardening by carburization on the fatigue behavior of AISI 316L austenitic stainless steel[J].Materials Science & Engineering, A, 2020.
[4] Egawa M , Ueda N , Nakata K ,et al.Effect of additive alloying element on plasma nitriding and carburizing behavior for austenitic stainless steels[J].Surface and Coatings Technology, 2010, 205.
2.The description of the morphology of the carburized layer is based on SEM photographs and has read a large amount of literatures. Based on the reviewers' comments, the high magnification SEM microstructure of the carburized layer has been added. We have also added metallographic photos of the carburized layer and the corresponding references for literature proof. The modifications are also marked in red.
3.Similar to low-temperature plasma nitriding technology, low-temperature plasma carburization technology is a sputtering-absorption-desorption-diffusion process. We use the schematic diagram of the formation of the nitriding layer (see Fig 1) to illustrate it[1]. In the glow ion diffusion furnace, under vacuum, the samples are used as the cathode, and a pulsed high-voltage current is applied to produce a glow discharge to ionize the carbon-containing medium (the carburizing gases are C2H2 and H2). The ionized carbon ions bombard the surface of the samples and penetrate into the surface while the samples are heated up, then continue to diffuse into the matrix. The carbon atoms that diffuse into the matrix will combine with the iron atoms inside the matrix in solid solution to form Sc phase. It can lead to the expansion and distortion of the matrix lattice structure. This process can obtain the carburized layer of a certain thickness, which contain the Sc phase mainly. It has been demonstrated by a large amount of relevant literatures that the C atoms enter the tetrahedral or octahedral gaps of the γ-Fe phase in a solid solution to form a metastable supersaturated solid solution Sc phase[2-5]. This phase has a disordered face-centered cubic lattice structure. We can also prove it with finite element or first principles, but this belongs to our future research target. Based on the reviewers' comments, we have added the corresponding references for literature proof. The modifications are also marked in red.
As a large number of carbon atoms will combine with iron atoms to form the Sc phase in the low-temperature plasma carburization process, there will not be a large number of individual carbon atoms left in the carburized layer. We generally clean and polish a very thin black film covering the carburized layer during the plasma carburizing process. We use the polished surface for testing and characterization, so there will not be any peaks corresponding to the carburized layer in the XRD pattern.
References:
[1] WANG Zilong. The Study on Low Temperature Plasma Nitriding of Several Stainless Steels [D].Lanzhou University of Technology, 2016.
[2] Liu H Y, Che H L, Li G B, et al. Low-pressure hollow cathode plasma source carburizing technique at low temperature[J].Surface and Coatings Technology, 2021, 422:127511.
[3] Marcos, Antnio. Wear resistance of AISI 304 stainless steel submitted to low temperature plasma carburizing.[J].REM - International Engineering Journal, 2017, 70(3):293-298.
[4] Sun Y. Tribocorrosion behavior of low temperature plasma carburized stainless steel[J].Surface & Coatings Technology, 2013, 228(228):S342-S348.
[5] Li Y, Li W, Zhu X, et al. Mechanism of improved hydrogen embrittlement resistance of low-temperature plasma carburised stainless steel[J].Surface Engineering, 2016:1-4.
4.The residual stress of the carburized layer can offset the applied stress to synthesize the total stress, as well as the effect on the fatigue strength, the residual stress offset the applied stress to a large extent. The residual stress of the carburized layer plays a major role in improving the fatigue strength, fatigue crack initiation, and extension resistance(see fig 2)[1]. In the contact state, if only the normal force exists, the three principal stresses of the stress field are compressive stresses. when there is friction, the two principal stresses can be changed into tensile stresses, although the existence of tensile stresses will promote the crack initiation and extension and produce unfavorable effects, but the existence of the residual stresses in the carburized layer can offset the unfavorable effects of the tensile stresses and reduce the chances of crack initiation and extension[2-4]. Due to the high surface hardness and brittleness of the carburized layer, it is mainly subject to local brittle fracture and microcracks in the fretting wear process, the fretting wear mechanism of the carburized layer is fatigue peeling and adhesion. The fretting wear mechanism of AISI 316L steel is severe plastic deformation and adhesion, because of its low hardness and high susceptibility to adhesion, which results in a large scale of cracking compared to that of the carburized layer. The size and number of microcracks of AISI 316L steel at wear interface are more than that of the carburized layer.
In addition, the local SEM images of the wear marks are generally selected to represent the typical damage morphologies at the contact interface and do not fully characterize the crack distribution at the contact interface.
References:
[1] Chen D H, Teng L X, Li G J, et al. Distribution Characteristics of Residual Stress in Carburized and Hardened Case[J].Heat Treatment, 2011, 26(2):65-71.
[2] Feng Z X, Zhang J Z, Chen X Z. On the Role of Compressive Residual Stress in Carburized Steels[J].Chinese Journal of Mechanical Engineering, 1990, 26(3):79-86.
[3] Liu Z, Wang S, Feng Y, et al. Residual stress relaxation in the carburized case of austenitic stainless steel under alternating loading[J].International Journal of Fatigue, 2022(159-):159.
[4]William E H, Eralp D, Dylan A, et al. The inclusion and role of micro mechanical residual stress on deformation of stainless steel type 316L at grain level[J].Materials Science Engineering A, 2023,876.
5.The coefficient of friction was related to many factors, such as the state of the contact surface (roughness, number and type of micro-convex body, etc.), the temperature of the contact interface, abrasive debris, and the external environment. The friction coefficient of GCr15/PC was slightly higher than that of GCr15/316L. There are four potential reasons that could explain this difference[1-3]: (1) Surface hardness. Abrasive chips participate in the wear process as a third body by scraping or ploughing the surface during fretting wear. The bearing capacity and anti-deformation capacity of these wear chips affect the friction coefficient. The influence of the relatively soft abrasive particles produces by the untreated AISI 316L steel on the friction coefficient is not as significant as that of the hard-phase abrasive particles produced by the carburized layer. (2) The "friction flashover temperature" occurrs at the wear interface during fretting wear. AISI 316L steel is easier to form an oxide film than a carburized layer as the friction temperature raises. (3) The crystal structure. Face-centered cubic austenite steel has a larger lattice than other crystal structures (BCC, HCP, etc.). The expansion distortion of the original austenite lattice after plasma carburizing mean that the carburized layer has a slightly larger friction coefficient than the untreated AISI 316L steel. (4) Surface roughness. The roughness of the carburized layer is slightly higher than that of the AISI 316L steel, the number of micro-convex body on the contact surface is higher, resulting in higher relative shear strength and elevated friction stress. After the initial contact, the micro-convex body of the carburized layer are gradually sheared by the upper specimen, and the wear chips fill the contact interface. When the number of wear chips are less than enough to fill the micro-convex body troughs, the tribological behaviors increases such as extrusion, ploughing, and grinding. The friction coefficient increases. However, when the number of wear chips continue to increase to fill the micro-convex body troughs, the abrasive layer (the third body layer) can be formed at the contact interface and the friction coefficient decreases slightly. The third body layer cab soon be extruded, ploughing, and grinding by the upper specimen to form new wear chips, and therefore the friction coefficient increases again. When the displacement is 50 μm, the transverse wear is very slight. The SEM images show that the wear of the carburized layer is very slight, resulting in a lower friction coefficient of the carburized layer than that of AISI 316L steel.
References:
[1] Sarangi C K, Sahu B P, Mishra B K, et al. Pulse electrodeposition and characterization of graphene oxide particle-reinforced Ni–W alloy matrix nanocomposite coatings[J].Journal of Applied Electrochemistry, 2020, 50(2):265-279.
[2] A Z H D, B W Z, B H W K, et al. Surface hardening of laser melting deposited 12CrNi2 alloy steel by enhanced plasma carburizing via hollow cathode discharge[J].Surface and Coatings Technology, 2020, 397.
[3] Hu, G.; Cai, X.; Rong, Y. Material Science Foundation[M]. Shanghai: SHANGHAI JIAO TONG UNIVERSITY PRESS, 2010.
6.Based on the reviewers' comments, we have made appropriate revisions to the abstract. Some old references in the abstract has been replaced. The structure of the article has been adjusted. The modifications are also marked in red.
7.Based on the reviewers' comments, we have modified the titles of the figures. The modifications are also marked in red.
8.Based on the reviewers' comments, we have modified fig.8. The modifications are also marked in red.

Round 2
Reviewer 2 Report
Comments and Suggestions for Authors
Now, the revised version is ready for acceptance.
Author Response
We are very grateful to the reviewer for the deep analysis and thinking of my article and provide some constructive suggestions and improvement plans. Thanks to the reviewer for finding and correcting some grammatical or typographical errors in my article, the overall quality has been improved. We are particularly thankful for the reviewer's positive recognition of our work, which serves as a motivating factor for us to continue our dedicated efforts. We are committed to further refining and advancing our research.
Once again, we sincerely appreciate the reviewer's expertise and time invested in evaluating our paper. Your contributions have been invaluable, and we are grateful for your support. We will continue to work hard.

Reviewer 3 Report
Comments and Suggestions for Authors
Well explained response letter.
Revisions seem sufficient for publication.
One comment is that there seems to be an issue with Figure 3. Might be a loading issue on my end, but I think it is lacking figure 3d. Please check.
Author Response
We feel sorry for our carelessness. Based on the reviewers' comments, we have
modified fig 3 in our resubmitted manuscript. We have added figure 3d. The
modifications are also marked in red.
